# Galvanic-Replacement-Assisted Synthesis of Nanostructured Silver-Surface for SERS Characterization of Two-Dimensional Polymers

**DOI:** 10.3390/s24020474

**Published:** 2024-01-12

**Authors:** Wenkai Zhao, Runxiang Tan, Yanping Yang, Haoyong Yang, Jianing Wang, Xiaodong Yin, Daheng Wu, Tao Zhang

**Affiliations:** 1Key Laboratory of Marine Materials and Related Technologies, Zhejiang Key Laboratory of Marine Materials and Protective Technologies, Ningbo Institute of Materials Technology and Engineering, Chinese Academy of Sciences, Ningbo 315201, China; 2University of Chinese Academy of Sciences, Beijing 100049, China; 3Key Laboratory of Leather Chemistry and Engineering of the Education Ministry, Sichuan University, Chengdu 610065, China

**Keywords:** galvanic-replacement-assisted synthesis, nanostructured silver-substrate, nanoAg@Cu, two-dimensional polymers, SERS spectroscopy

## Abstract

Surface-enhanced Raman scattering (SERS) spectroscopy is a powerful technology in trace analysis. However, the wide applications of SERS in practice are limited by the expensive substrate materials and the complicated preparation processes. Here we report a simple and economical galvanic-replacement-assisted synthesis route to prepare Ag nanoparticles on Cu(0) foil (nanoAg@Cu), which can be directly used as SERS substrate. The fabrication process is fast (ca. 10 min) and easily scaled up to centimeters or even larger. In addition, the morphology of the nanoAg@Cu (with Ag particles size from 30 nm to 160 nm) can be adjusted by various additives (e.g., amino-containing ligands). Finally, we show that the as-prepared nanoAg@Cu can be used for SERS characterization of two-dimensional polymers, and ca. 298 times relative enhancement of Raman intensity is achieved. This work offers a simple and economical strategy for the scalable fabrication of silver-based SERS substrate in thin film analysis.

## 1. Introduction

Two-dimensional polymers (2DPs) are a class of organic two-dimensional materials comprising free-standing, single-atom/monomer-thick, planar, covalent networks with well-defined periodicity along two orthogonal directions [1,2,3,4,5]. Taking the advantages of high specific surface area, excellent charge carrier mobility and superior electrical conductivity [6,7], 2DPs have potential applications in many fields, such as gas and ion separation [8,9,10], capacitive energy storage [11,12,13,14,15], chemical sensing [16,17,18], etc. The performances in these applications highly rely on the structures of the 2DPs. Nevertheless, the structural characterization at the molecular or atomic level is the greatest challenge in this field [1,19]. Raman is an important technique to characterize the bonding information of 2DPs and further used to analyze the molecular structure. However, the extremely thin feature of 2DPs makes it difficult to obtain a strong Raman signal. Recently, some researchers [20,21,22,23,24] use tip-enhanced Raman spectroscopy (TERS) to monitor the formation of 2DPs. Nonetheless, TERS requires sophisticated equipment that is not assessable widely. In contrast, surface-enhanced Raman scattering spectroscopy (SERS) is a more economical and convenient method.

SERS, as an emerging technology with the ability to analyze chemical characteristics quickly and sensitively, has been widely used in the research fields of chemistry [25,26,27,28], biomedicine [29], food [30], environment [31], etc. These applications depend primarily on the unique features of the nanomaterials with distinctive component, structure, and assembly form [32,33,34,35,36,37,38,39]. The most researched SERS materials are Au and Ag nanoparticles because their assemblies have an outstanding SERS enhancement effect attributed to their abundant localized surface plasmons [40,41,42,43]. However, the applications of SERS in practice are limited by the high cost of substrate materials and the complicated preparation processes. Thus, exploiting an economical, convenient, and sensitive substrate is necessary to promote the application of SERS.

In this work, we show a facile and economical galvanic-replacement-assisted approach for the preparation of nanostructured Ag on Cu(0) surface, which can be directly used as SERS substrate for the characterization of 2DPs. The preparation process takes less than 10 min, and the morphology of the Ag surface can be adjusted by various amino-containing (-NH_2_) ligands. For example, the Ag nanoparticles with an average diameter ca. 80 nm can be obtained from 10 mM AgNO_3_ with the adjusting of 1 mM amino-terminated polyethylene glycol (HS-PEG-NH_2_), which, as SERS substrate for 2DPs’ characterization, give a relative Raman intensity enhancement of 298 times. This work offers a fast, economical, and convenient way for the fabrication of sensitive SERS substrate, which could facilitate the SERS technology in different application fields.

## 2. Materials and Methods

### 2.1. Materials

Cu(0) foil (with thickness of 0.05 mm) was purchased from Zhong Nuo New Materials (Beijing) Technology Co., Ltd. (Beijing, China). Hydrochloric acid (HCl, 37 wt % in water) and silver nitrate (AgNO_3_, >99.0%) were purchased from Si-nopharm Chemical (Shanghai, China). Diethylamine (DEA), 2-Mercaptoethylamine (MEA), Ytterbium (III) triflromthanesufonate hydrate (Yb(OTf)_3_), and sodium do-decyl benzene sulfona were bought from Aladdin (Shanghai, China). Amino-terminated polyethylene glycol (NH2-PEG-NH_2_ and HS-PEG-NH_2_, Mw = 2000 g/mol) were bought from J&K Chemicals (Shanghai, China). 5,10,15,20-tetrakis (4-aminophenyl) porphyrin and 2,5-dihydroxyterephthalaldehyde were purchased from Macklin (Shanghai, China). All aqueous solutions were prepared using deionized water.

### 2.2. The Fabrication of NanoAg@Cu

All aqueous solutions were prepared using deionized water. Firstly, the Cu(0) foils (1 cm × 1 cm) were immersed in 1 M HCl to remove the oxide at the surface; 1 mL mixture of AgNO_3_ solution (1 mM) and -NH_2_ ligands solution (1 mM) was loaded into the 24-well plate with a diameter of 1 cm for each hole. Then the soaked Cu(0) foils were washed 2 or 3 times with deionized water, and then the moisture on the surface was carefully absorbed with filter paper. Subsequently, the treated Cu(0) foil was immersed into the mixture solution and sonicated immediately for 2 min. Then, it was let to stand for 8 min at room temperature about 30–35 °C. The obtained substrates were washed with deionized water and gently absorbed with filter paper to remove the excess water on the surface. Lastly, the substrates were put into a 60 °C oven to dry for 1 h for further use.

### 2.3. NanoAg@Cu Used for the SERS Detection of 2DPs

The 2DPs were fabricated according to the literature [1]. In brief, 20 μL sodium dodecyl benzene sulfona (1 mg mL^−1^ in chloroform) was first spread at the air–water interface formed by 50 mL deionized water in a 60 mL crystallization dish. After the evaporation of solvent for about 30 min, an aqueous solution of 0.5 mL 5,10,15,20-tetrakis (4-aminophenyl) porphyrin (1 mg mL^−1^ in 0.12 M HCl solution) was injected into the water phase at 50 °C and dispersed for 20 min. Then 0.5 mL 2,5-dihydroxyterephthaladhyde monomer (2.8 µmole) was added to the water phase and diffused for 20 min. Lastly, 2 mL 0.001 mmol Yb(OTf)_3_ was added to the water phase. The reaction was kept at 50 °C for several days to obtain 2DPs with different thicknesses.

After the preparation of the membrane, the film was gently taken with nanoAg@Cu substrate to make the substrate attach a layer of 2DPs evenly. Then the complex of nanoAg@Cu and 2DPs dried naturally at room temperature. Lastly, the complex was detected by Raman spectrum.

### 2.4. Characterization

The ultraviolet–visible (UV–vis) absorption spectra of nanoAg@Cu were recorded from an external visible near infrared spectrophotometer (LAMBDA, Perkin-Elmer, Waltham, MA, USA). Scanning electron microscope (SEM) images were performed with a field-emission scanning electron microscope (S-4800, Hitachi, Tokyo, Japan) and cold field high resolution scanning electron microscope (Regulus8230) at an acceleration voltage of 8 kV. The element mapping images were obtained from Regulus8230 at an acceleration voltage of 15 kV and current of 10 µA. Size distributions were determined from SEM images using digital micrograph software and more than 20 nanoparticles for each sample were measured. Microscopic images were recorded from fluorescence metallographic microscope (NMM-800RF, Ningbo Op Instrument Co., Ltd. (Ningbo, China)). Scanning Probe Microscope (SPM) images were recorded from Dimension 3100 (Vec-co, New York, NY, USA). X-ray diffraction (XRD) were detected with X-ray powder diffractometer (D8 ADVANCE DAVINCI, German BRUKER, Karlsruhe, Germany). Raman spectrums were obtained from Confocal Raman Microscope (RENISHAW, Renishaw inVia Reflex, New Mills, UK) with excitation light of 532 nm; the power density of the laser was set to 10% or 1% and exposure time was set to 1 s.

## 3. Results and Discussion

### 3.1. The Fabrication and Characterization of NanoAg@Cu

Previously, our group and others showed that Cu(0) surface as an abundant Cu source can be used to synthesize functional polymer materials owing to the generation of nascent Cu^0^ or Cu^I/II^ species as activators or catalysts in polar liquids or alkaline solutions [44,45,46,47,48]. In addition to these emerging reaction processes, the instinctive oxidation–reduction of metal with different redox potential is also a basic reaction. As the standard electrode potential of the Ag^+^/Ag pair (E°*_Ag_^+^*_/*Ag*_ = +0.799 V) is higher than that of the Cu^2+^/Cu pair (E°*_Cu_^2+^*_/*Cu*_ = +0.337 V), the oxidation–reduction reaction take place when Ag^+^ approaches to the Cu(0) surface. The Ag^+^ is reduced by gaining one electron and Cu is oxidized by losing two electrons, which is the basic reaction for the formation of nanoAg@Cu.

Figure 1 illustrates the fabrication process of nanoAg@Cu. After immersing the Cu(0) foil in AgNO_3_ solution, the reaction vessel was put in an ultrasound device immediately with ultrasound for 2 min. Then the reaction vessel stood still for 8 min so that the nanoAg@Cu would be obtained (Figure 1a). In this process, the morphology of nanoAg@Cu would be influenced by various factors. Firstly, lattice mismatch is an important factor for epitaxial growth of Ag on Cu(0) surface. With the help of redox reaction, Ag^+^ was reduced to Ag atom when it approached the Cu(0) surface (Figure 1(bI)). The formed Ag atoms on Cu(0) surface were not divorced which attributed to the existence of a metal bond between Ag and Cu. These initially formed Ag atoms acted as crystal nuclei [49], while the subsequently formed Ag atoms acted as epitaxial growth materials (Figure 1(bII)). For the epitaxial growth of binary metal, three growth modes including Frank–van der Merwe (F–M) mode, Volmer–Webber (V–W) mode, and Stranski–Krastanow (S–K) mode existed, which correspond to three morphologies including concentric, eccentric, and island shape [50]. Island growth has been mostly limited to a few binary metal and semiconductor systems that involve materials with large lattice mismatches [51,52], The lattice mismatch of Ag and Cu is large, which leads the epitaxial growth of Ag on Cu(0) surface to form the S–K mode and generate an island shape (Figure 1(bIII)). Lastly, these island structures exhibit as relatively independent nanoparticles on Cu(0) surface. As Figure 1c,d show, the nano-topography and element distribution of the front side and cross-section of nanoAg@Cu are obtained from one designed condition. In further measurements and calculations, the average diameters of relatively independent nanoparticles in Figure 1c are around 55 nm (Figure 1e).

Besides the effects of lattice mismatch, nucleation and growth rate can also affect the formation of nanoAg@Cu [49,53]. More uniform structures were obtained while introducing the ultrasound process, which was credited to the increased nucleation rate and more uniform distribution of crystal core induced by ultrasound (Appendix A). The nucleation and growth process also are affected by the adding rate of AgNO_3_ solution (Appendix A). By reducing the adding rate of AgNO_3_ solution, some more independent Ag islands appeared at the Cu(0) surface. The big gap of Ag nanoparticles was not conducive to the uniformity of the detection signal, so the one-time joining of AgNO_3_ solution was selected.

In addition to the conventional control factors, the emergence of ligands brings more possibilities for the growth of nanocrystals [54,55]. The slight changes of pH and functional group will impact the process of crystals [56]. In this system, we selected a few of the -NH_2_ ligands with weakly basic functional groups to affect the growth of Ag. Taking the two more typical results to illustrate, the SEM image of nanoAg@Cu adjusted by little molecules of diethylamine (DEA) shows that lots of larger sheet structures were produced along with the agglomeration of some small particles (Appendix A). Especially, the SEM image of nanoAg@Cu adjusted by HS-PEG-NH_2_ shows that more rounded and regularly distributed Ag nanoparticles with average diameters ca. 120 nm were produced (Figure 2a,b). This obvious morphological difference is mainly caused by the different role of ligands. The X-ray photoelectron spectroscopy of the nanoAg@Cu tuned by HS-PEG-NH_2_ shows a stronger signal of Cu 2p at around 932.8 eV and Ag 3d at 368.6 and 374.6 eV (Figure 2c and Appendix A). The X-ray powder diffractometer (XRD) characteristic is weak and the little peak at 2-theta of 38° revealed that the existence of Ag in nanoAg@Cu is regulated by HS-PEG-NH_2_ (Appendix A). However, the XRD shows almost no peak of Ag obtained from nanoAg@Cu tuned by DEA. The UV–vis absorption spectrum shows that the absorption peak of nanoAg@Cu is mainly at about 300 nm which has a slight blue shift compared to the peak of Cu(0) foil (Figure 2d).

### 3.2. The NanoAg@Cu Used for Detection of 2DPs

Distinctive crystallized imine-based 2DPs were prepared by interfacial synthesis according to the literature report (Figure 3a) [57]. The obtained 2DPs were characterized by fluorescence metallographic microscope, scanning probe microscope (SPM), and confocal Raman microscope. These revealed that the 2DPs with thicknesses of about 20 nm were relatively even (Figure 3b,c). The Raman spectrum shows the characteristics of crystalline 2DPs on silicon (Appendix A). However, the intensity of the obtained peaks is so low that the signal is easily disturbed by noise. To prove the enhanced properties of nanoAg@Cu for the detection of 2DPs, the obtained nanoAg@Cu were combined with 2DPs and used for further characterization (Figure 3d). The laser excitation wavelength of 532 nm, power intensity of 10%, which is equivalent to 1.4 mW, and exposure time of 1 s were set as the Raman monitoring parameters. Then a series of nanoAg@Cu were used to enhance the signals of 2DPs. After transferring the 2DPs to nanoAg@Cu and drying at room temperature, the complexes were examined by Raman spectrometer (Figure 3e). The Raman spectra show the N-H bending band at 1280 cm^−1^ and the aldehyde C=O stretch at 1675 cm^−1^ serve as confirmation for the synthesis of 2DP materials. To compare the SERS results with different laboratories worldwide, the most popular approach is to use an enhancement factor as a measure of the plasmonic enhancement, which is related to the affinity of an analyte to the surface of nanoparticles [58,59]. However, thin film samples in a two-dimensional state cannot uniformly adsorb onto the surface of plasma particles, therefore, estimating the relative SERS signal enhancement is a more feasible means. The result shows that the nanoAg@Cu obtained from pure AgNO_3_ solutions has a high enhancement of intensity and the Ag-substrate adjusted by regulatory ligands process varying degrees of enhancement. Specifically, the enhancement effect of nanoAg@Cu adjusted by DEA and HS-PEG-NH_2_ is close and much higher than that adjusted by mercaptoethylamine (MEA) (Figure 3f).

The nanoAg@Cu regulated by HS-PEG-NH_2_ was used for further research since it possesses a more regular morphology and exhibits a higher enhancement. An obvious enhancement trend is shown with the increase of the concentration of AgNO_3_ solution (Figure 4a), and an enhancement of Raman intensity up to 298 times was achieved (Figure 4b). When the power density of the laser is set to 10%, which is equivalent to 1.4 mW, the SERS signals from the substrate prepared by 0.5 mM and 1 mM AgNO_3_ solution show different degrees of enhancement, but the SERS signals from the substrate prepared by 5 mM AgNO_3_ solution exceeded the detection limit (Appendix A). When the power density of the laser is changed to 1%, which is equivalent to 0.14 mW, the signals exhibit a clear peak shape with a large enhancement (Appendix A). The great change may be caused by the microtopography. When increasing the concentration of AgNO_3_ from 1 mM to 10 mM, the obtained Ag substrate exhibits different morphology (Appendix A). For specific performance, Ag mainly dispersed on the Cu(0) surface in the form of little nanoparticles when the concentration of AgNO_3_ is 1 mM (Appendix A). However, Ag dispersed on the Cu(0) surface in the form of agglomerates of small nanoparticles when the concentration of AgNO_3_ is up to 5 mM and 10 mM, which may be owing to the sufficient AgNO_3_ solution, and are conducive to rapid and massive nucleation and quick growth of particles [53] (Appendix A). With measurement and calculation, the small particles in this case are around 30 nm, the size of agglomerates is in the range of 50 to 120 nm, and more are around 80 nm (Appendix A). When the concentration of AgNO_3_ is set to 10 mM, the aggregation behavior of small particles is no longer obvious and the obtained particles is about 100 nm, which is larger than the previous. Increasing the concentration of AgNO_3_ solution leads to an increase in the content of silver components and a tighter distribution of Ag nanoparticles, resulting in more hot spots, which will be beneficial to the increase of the Raman signal.

In the SERS measurements, the reproducibility is a significant issue. Using HS-PEG-NH_2_ as regulatory ligand, the test results showed a good repeatability while the concentration of AgNO_3_ solution is 5 mM (Figure 4c) but a weaker repeatability while the concentration of AgNO_3_ solution is up to 10 mM (Appendix A). Therefore, the concentration of 5 mM is more conducive to the enhancement under this condition.

In order to further investigate the detection performance of the obtained nanoAg@Cu, in our preliminary research, the 2DPs with different thicknesses and layers were characterized (Figure 5). After the fabrication of 2DPs with thicknesses of 3 nm and 11 nm (Figure 5a–d), and the preparation of nanoAg@Cu substrates from 5 mM AgNO_3_ solution adjusted by HS-PEG-NH_2_, the complex of 2DPs and the nanoAg@Cu were characterized by Raman spectroscopy. For the test parameters, the power density was set to 10% and exposure time was set to 1 s with excitation light of 532 nm. The results show that with the increase in thickness or layers of the 2DPs, the SERS signals all increased and the peak shape was in accord with the Raman scattering characteristics of the crystalline 2DPs (Figure 5e,f). Therefore, it is very promising to correlate the detection signal with the film thickness and layer number information.

## 4. Conclusions

We report a facile and low-cost galvanic-replacement-assisted synthesis approach for the fabrication of nanoAg@Cu SERS substrates for the characterization of 2DPs. The fabrication process of nanoAg@Cu is fast (ca. 10 min) and can be easily scaled up to centimeters. During the fabrication of nanoAg@Cu, ultrasound is beneficial to gain the more uniform nucleation and growth of Ag on the Cu(0) surface. In addition, the growth process can be adjusted by -NH_2_ ligands. Adjusting by HS-PEG-NH_2_ ligand, Ag nanoparticles with a diameter size from 30 to 160 nm can be obtained on the Cu(0) surface. The obtained nanoAg@Cu were used directly as SERS substrate for the analysis of imine-based 2DPs and an enhancement of Raman intensity up to 298 times was achieved. This study may shed light on the fabrication of economical, convenient, and sensitive SERS substrate and promote the application of SERS substrate in thin film analysis.

## Figures and Tables

**Figure 1 sensors-24-00474-f001:**
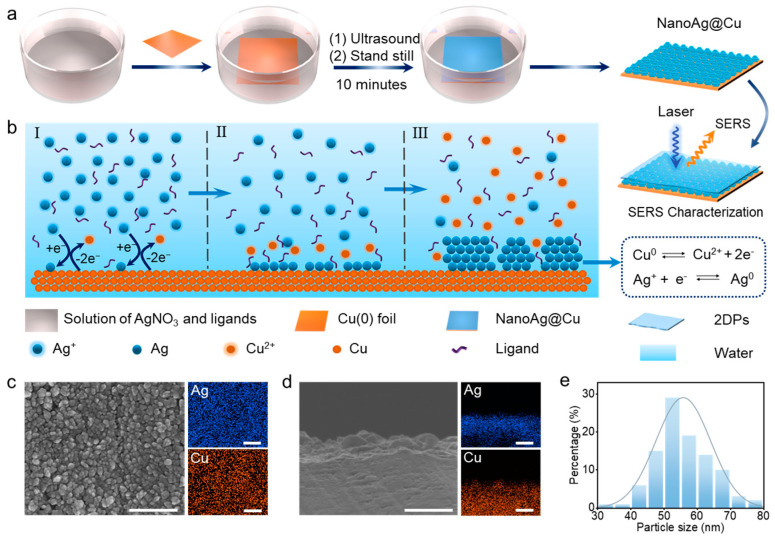
Schematic diagram of preparation of nanoAg@Cu. The macro (**a**) and microscopic (**b**) schematic diagram of nanoAg@Cu substrate fabrication process. Ag+ was reduced to Ag atom when it approached the Cu(0) surface (I), subsequently formed Ag atoms acted as epitaxial growth materials (II) and the epitaxial growth of Ag on Cu(0) surface to form the S–K mode and generate an island shape (III). The SEM and corresponding element mapping images of the front side (**c**) and cross-section (**d**) of the nanoAg@Cu from one designed condition, scale bar: 500 nm. (**e**) Distribution of particle size in (**c**).

**Figure 2 sensors-24-00474-f002:**
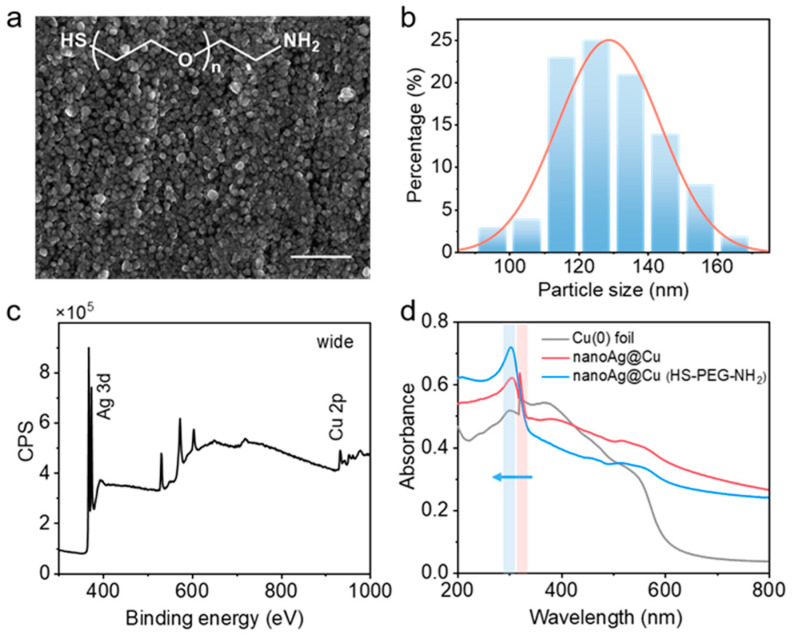
Characteristics of nanoAg@Cu adjusted by HS-PEG-NH_2_. (**a**) SEM image. (**b**) Distribution of particle size in (**a**). (**c**) X-ray photoelectron spectroscopy. (**d**) UV–vis–VIR absorption spectra. Scale bar: 1 μm.

**Figure 3 sensors-24-00474-f003:**
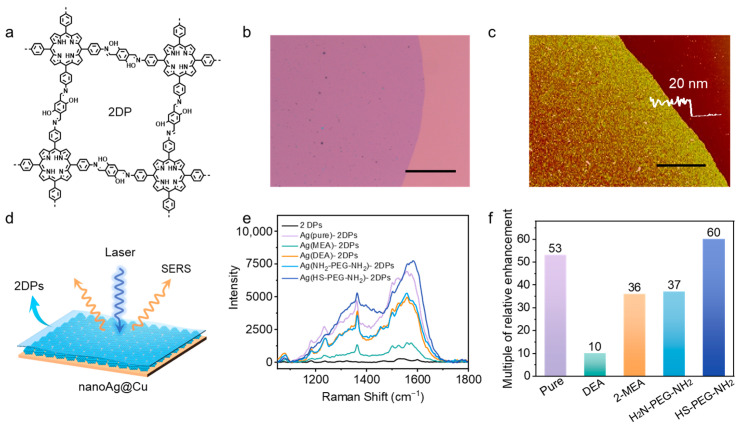
Effects of the nanoAg@Cu substrates derived from different -NH_2_ ligands on SERS signal. (**a**) Chemical structure of synthetic 2DP. (**b**) Microscopic image of 2DP. Scale bar: 100 μm. (**c**) SPM image of 2DP. Scale bar: 10 μm. (**d**) Schematic diagram of nanoAg@Cu complexed with 2DPs used for characteristics of Raman spectrum. (**e**) SERS spectrum of 2DPs from substrates regulated by the -NH_2_ ligands. The -NH_2_ ligands are diethylamine (DEA), 2-Mercaptoethylamine (MEA), H_2_N-PEG-NH_2_, and HS-PEG-NH_2_, respectively. (**f**) Comparison chart of relative integrated area of Raman signal intensity in (**e**).

**Figure 4 sensors-24-00474-f004:**
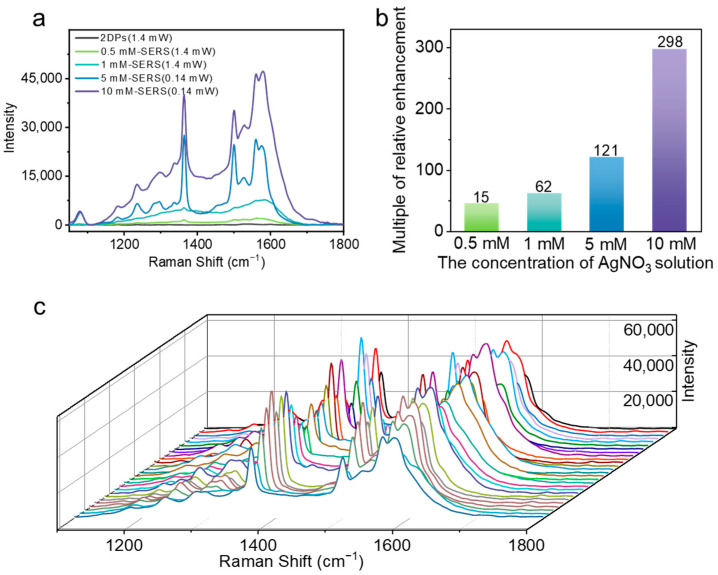
Effects of the nanoAg@Cu substrates derived from different concentrations of AgNO3 on SERS signal and reproducibility characteristics. (**a**) SERS spectrum of 2DPs from substrates obtained by changing the concentration of AgNO3 solution with 0.5 mM, 1 mM, 5 mM, and 10 mM. (**b**) Comparison chart of integrated area of Raman signal intensity in (**a**). (**c**) The repeatability test results of the SERS signals from 30 points of nanoAg@Cu obtained from 5 mM AgNO3 solution.

**Figure 5 sensors-24-00474-f005:**
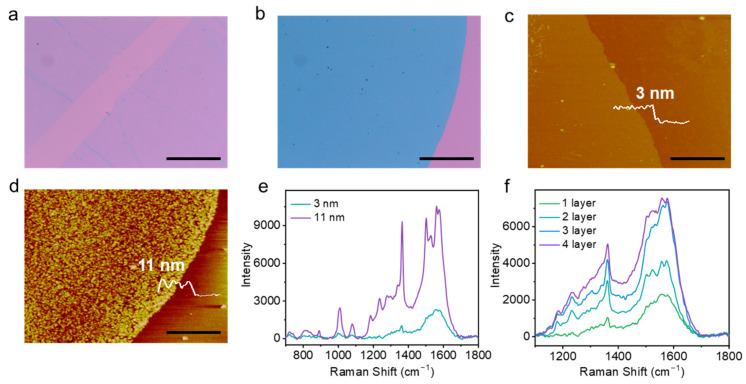
Characterization of 2DPs with different thicknesses. Microscopic images of 2DPs with thicknesses of 3 nm (**a**) and 11 nm (**b**). SPM images of 2DPs with thicknesses of 3 nm (**c**) and 11 nm (**d**). SERS signals of 2DP with different thicknesses (**e**) and layers (**f**). The scale bar in (**a**,**b**) is 100 μm. The scale bar in (**c**,**d**) is 1 μm.

## Data Availability

Data are contained within the article.

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
