# Peer review of "Galvanic-Replacement-Assisted Synthesis of Nanostructured Silver-Surface for SERS Characterization of Two-Dimensional Polymers"

_sensors, 2024, doi:10.3390/s24020474_

Round 1
Reviewer 1 Report
Comments and Suggestions for Authors
The manuscript “Galvanic-Replacement-Assisted Synthesis of Nanostructured Silver-Surface for SERS Characterization of Two-Dimensional Polymers” by Wenkai Zhao, Runxiang Tan, Yanping Yang, Haoyong Yang, Jianing Wang, Xiaodong Yin, Daheng Wu and Tao Zhang investigates a simple and economical galvanic-replacement-assisted synthesis route to prepare Ag nanoparticles on Cu foil, which can be directly used as SERS substrate. The prepared SERS substrates with Ag particles size from 30 nm to 160 nm were used for SERS characterization of 2D polymers with 10^2 times relative enhancement.
The work describes in detail the process of manufacturing of SERS substrates. The ultraviolet-visible absorption spectra clearly demonstrate the excitation of plasmon resonances on silver nanoparticles which lead to EM field enhancement and SERS. The main advantage of the proposed SERS substrates is the fast fabrication (10 minutes) and easily scalable SERS-active surface. The article is well read, the illustrative materials are convincing.
However, this manuscript did not discuss several points:
1) What maximum level of SERS signal has been achieved in similar Ag-substrates in other works?
2) It would be useful to update the bibliography, because there are no works for the last 3 years.
3) What Raman shift was chosen to determine the detection limit and what vibration does it correspond to?
The manuscript can be considered for publishing in Sensors, and I support publication with minor edits.
Author Response
Thank you for your criticism and correction of our work, the revisions are in the annex.

Reviewer 2 Report
Comments and Suggestions for Authors
This manuscript concerns about the fabrication of Ag nanoparticles on Cu(0) foil (nanoAg@Cu) by a galvanic-replacement-assisted synthesis route, which are examined as SERS substrate for the detection of two-dimensional polymers, and ca. 298 times relative enhancement of Raman intensity is achieved. The synthetic route is suggested to be fast (ca. 10 min) and easily scaled up, while the extent of enhancement of the substrates is not so attractive, and there are many flaws exist in the manuscript which prevent it to be suitable for publication. Here are some comments.
(1) The SERS enhancement obtained with the nanoAg@Cu should be compared with other documented substrates for the detection of 2D polymers, in order to illustrate the advantages achieved in this work.
(2) Line 183, a blue shift is observed from the UV-vis absorption spectrum of nanoAg@Cu, when compared to that of Cu(0) foil (Figure 2d). The authors should give a brief illustration about this phenomenon. Is it related to the enhanced SERS performance?
(3) Whether the variation in roughness of the substrates contributes to the different SERS enhancements?
(4) The microscopic images of 2DPs such as Figure 3b, Figure 5a, 5b are meaningless, at least additional descriptions about the images should be given in the context.
(5) Raman signals attributable to the 2D polymers should be marked and described.
(6) In Figure 4a, for a reasonable comparison, the Raman profiles should be obtained under the same power density of the laser.
(7) In Figure 5c and 5d, 2DPs with thicknesses of 3 nm and 11 nm exhibit obviously different roughness, are they contributing to the observed variations in SERS signals?
(8) What’s the relationship between the thickness and layers of the 2DPs?
(9) It is suggested one can correlate the detection signal with the film thickness and layer number information, can they be quantitatively correlated?
(10) The English in the manuscript should be thoroughly improved, since it cause certain confusions in understanding the results. For example, Line 209, “However, the film with lots of 2D bindings can’t adsorb on the surface of plasma particles uniformly, therefore estimate the relative SERS signal enhancement is a more feasible means”. Line 238, “conducive to rapid and massive nucleation and quick growth of particles46”. Line 242, “which are larger than the previous”.
Comments on the Quality of English LanguageThe English in the manuscript should be thoroughly improved, since it cause certain confusions in understanding the results.
Author Response

(The authors gave the same response as above.)

Reviewer 3 Report
Comments and Suggestions for Authors
This manuscript reported a galvanic-replacement-assisted synthesis route to prepare Ag nanoparticles on Cu(0) foil which can be used as the plasmonic substrate for SERS characterization of 2D polymers. The fabrication process is easy and scalable, and the size and morphology of Ag nanoparticles on Cu foil and hence the SERS enhancement factor can be modulated by parameters such as the concentration of AgNO3 and additives. Preliminary results obtained from the nanoAg@Cu substrate prepared in this method showed that the Raman signal intensity scales with the polymer thickness/layer numbers, making it a promising candidate for (semi-)quantitative analysis applications. Overall, this manuscript is clear, relevant for the field and the conclusions are consistent with the arguments presented. Would recommend to accept this manuscript after the following questions are addressed—
1. Since the electronic and surface plasmon bands of the substrate is crucial to the Raman scattering enhancement, a more detailed discussion of the absorption spectra (figure 2d) should be included. What is the sharp peak at ~300 nm? How to understand the blue shift of the peak when Ag nanoparticles are grown at the Cu surface? Also, the spectra of both nanoAg@Cu and nanoAg@Cu(HS-PEG-NH2) show a very broad absorption band extending beyond 600 nm. Is this abortion band real or due to measurement artifact? If this band is real, can it be ascribed to the plasmon mode of the aggregated Ag nanoparticles?
2. In section 3.2 the authors proposed the higher enhancement factor observed on the substrate obtained from 10 mM AgNO3 concentration than the substrate made from 5 mM can be explained by more plasmonic hot spots resulted from the denser distribution of Ag nanoparticles. However, the agglomeration of Ag nanoparticles is much more obvious on the substrate made from 5mM AgNO3 concentration based on the data. Don’t agglomerated/clustered Ag nanoparticles provide more hot spots? Does the morphology of Ag nanoparticles also impact how closely and uniformly the 2D polymer specimen can attach to the plasmonic surface and in this way influence the enhancement factor and signal reproductivity?
Author Response

(The authors gave the same response as above.)

Round 2
Reviewer 2 Report
Comments and Suggestions for Authors
The authors have responded to the comments.
Comments on the Quality of English LanguageIt's suggested to be further improved.